# Alzheimer’s Disease and Type 2 Diabetes Mellitus: The Use of MCT Oil and a Ketogenic Diet

**DOI:** 10.3390/ijms222212310

**Published:** 2021-11-15

**Authors:** Junpei Takeishi, Yasuko Tatewaki, Taizen Nakase, Yumi Takano, Naoki Tomita, Shuzo Yamamoto, Tatsushi Mutoh, Yasuyuki Taki

**Affiliations:** 1Department of Aging Research and Geriatric Medicine, Institute of Development, Aging and Cancer, Tohoku University, Sendai 980-8575, Japan; junpei.takeishi.s5@dc.tohoku.ac.jp (J.T.); yumi.takano.b7@tohoku.ac.jp (Y.T.); naoki.tomita.b1@tohoku.ac.jp (N.T.); shuzo.yamamoto.c7@tohoku.ac.jp (S.Y.); tmutoh@tohoku.ac.jp (T.M.); yasuyuki.taki.c7@tohoku.ac.jp (Y.T.); 2Department of Geriatric Medicine and Neuroimaging, Tohoku University Hospital, Sendai 980-8575, Japan; 3Smart Aging Research Center, Tohoku University, Sendai 980-8575, Japan

**Keywords:** Alzheimer’s disease, amyloid-beta, type 2 diabetes mellitus, insulin resistance, glucose metabolism, ketone metabolism, ketogenic diet, coconut oil, MCT oil

## Abstract

Recently, type 2 diabetes mellitus (T2DM) has been reported to be strongly associated with Alzheimer’s disease (AD). This is partly due to insulin resistance in the brain. Insulin signaling and the number of insulin receptors may decline in the brain of T2DM patients, resulting in impaired synaptic formation, neuronal plasticity, and mitochondrial metabolism. In AD patients, hypometabolism of glucose in the brain is observed before the onset of symptoms. Amyloid-β accumulation, a main pathology of AD, also relates to impaired insulin action and glucose metabolism, although ketone metabolism is not affected. Therefore, the shift from glucose metabolism to ketone metabolism may be a reasonable pathway for neuronal protection. To promote ketone metabolism, medium-chain triglyceride (MCT) oil and a ketogenic diet could be introduced as an alternative source of energy in the brain of AD patients.

## 1. Introduction

Today, as many as 50 million people worldwide have dementia, the most common form of which is Alzheimer’s disease (AD). AD typically manifests as a progressive loss of memory and cognitive function. Cerebral plaques laden with amyloid-β (Aβ) and intracellular neurofibrillary tangles composed of tau are important hallmarks of AD [1]. Aβ accumulation is associated with functional and structural brain alterations, consistent with the patterns of abnormalities seen in patients with mild cognitive impairment (MCI) as well as AD [2]. The amyloid cascade hypothesis suggests that the deposition of Aβ triggers neuronal dysfunction, vascular damage, and cell death in the brain [3]. Moreover, Aβ has been reported to directly impair the glycolytic and tricarboxylic acid (TCA) pathway [1]. In fact, patients with AD show insulin resistance in the brain [4]. 

Regarding glucose metabolism disorders, diabetes mellitus (DM) is one of the most common public health problems worldwide. The global prevalence of DM in 2019 was estimated to be 9.3% (463 million people), rising to 10.2% (578 million) by 2030 and 10.9% (700 million) by 2045 [5]. The pathogenesis of type 2 DM (T2DM) is primarily initiated by the inadequate function of pancreatic β-cells in response to glycemic overload, which then causes insulin resistance. Thus, patients with T2DM are relatively insulin deficient.

Herein, it can be said that insulin plays an important role in the mechanisms involved in the pathophysiological hypometabolism of glucose in both AD and T2DM. In fact, T2DM was reported to predispose patients to neurodegenerative disorders, including AD [6]. A meta-analysis reported that the odds ratio for conversion from MCI into AD in patients with and without diabetes was 1.65 (95% CI 1.12 to 2.43) [7]. Therefore, in this review, we will focus on the shared pathogeneses in AD and T2DM and the decreased use of glucose in the brain. To this end, for a potential of nonpharmacological molecular biology treatment, we will discuss the ketogenic diet (KD), a high-fat and low-carbohydrate diet in which coconut oil and medium-chain triglyceride (MCT) oil are often used as an alternative source of energy against glucose. 

## 2. Risk of AD

The risk of developing AD is divided into two categories: inherited and modifiable.

Genetic involvement is an inherited risk factor for AD. Mutations in the genes of amyloid precursor protein (APP), presenilin 1 (PSEN1), and PSEN2 are associated with the early onset of familial AD [8]. APP is normally cleaved by α-secretase, but its mutation affects α-cleavage and increases β- and γ-cleavage, resulting in the accumulation of Aβ. PSEN1 and PSEN2 are components of γ-secretase, and their mutation influences the increased activity of γ-secretase, resulting in the accumulation of Aβ. Additionally, a polymorphism in the apolipoprotein E (APOE) gene can be a major risk factor for sporadic AD. APOE has three isoforms: E2, E3, and E4. Thus, APOE4 is thought to increase the Aβ burden by interfering with Aβ clearance [9].

Numerous modifiable factors have been investigated as risk factors for AD [10]. Among them, lifestyle habits, such as decreased physical activity, midlife obesity, alcohol intake, and smoking, are important since they can be controlled in our daily life [11]. Moreover, such lifestyle habits are also risks of cerebrovascular disease [12], hypertension [13], dyslipidemia, and DM [14]. In fact, these diseases are involved in AD pathogenesis. Therefore, it can be said that we can partly control the onset of AD by changing such modifiable risk factors. 

## 3. AD and DM: Mechanisms of Cognitive Decline Associated with Insulin Resistance

Insulin is secreted from the pancreas, and regulates glucose metabolism by means of activation of the insulin receptor (IR). In patients with T2DM, because of the inappropriate secretion of insulin triggered by an abnormal glucose overload, insulin resistance will primarily be observed. Then, the persistent stress of insulin secretion may result in the dysfunction of pancreatic β-cells. Meanwhile, insulin has long been implicated in cognitive performance [15,16]. First, IRs are located in the central nervous system synapses [17,18], and insulin, which comes from the general circulation, can cross the blood–brain barrier (BBB) [19] via a saturable transporter [20]. Then, as shown in Figure 1, insulin binds to the IR at the synapse, and autophosphorylation of the IR occurs. An activated IR phosphorylates the insulin receptor substrate (IRS). This results in the cascade of phosphoinositide 3-kinase (PI3K) activation, phosphoinositide-dependent protein kinase-1 (PDK-1) activation, and protein kinase B (Akt) activation. Activated Akt can activate glycogen synthase kinase 3β (GSK3β), mammalian target of rapamycin complex 1 (mTORC1), and forkhead box O (FOXO1). Then, these downstream cascades influence the activity of AMPA-type glutamate receptor subunit-1 (GluA1) and NMDA-type glutamate receptor subunit 2 B (GluN2B) [21,22,23]. Since both GluA1 and GluN2B play a critical role in synaptic plasticity, insulin can be associated with memory and learning in the hippocampus [24]. Moreover, activated Akt promotes the phosphorylation of AMP-activated protein kinase (AMPK), leading to the activation of Sirtuin 1 (SIRT1) and peroxisome-proliferator-activated receptor γ co-activator 1α (PGC1α). This pathway plays an important role in mitochondrial metabolism, a major source of ATP [25,26,27]. 

Similar to the insulin pathology in T2DM patients, it has been reported that impaired insulin action is observed in the brain of AD patients [4]. Three pathologies have been suspected: reduced transport of insulin into the brain [28,29], reduced insulin levels in the brain [30], and poorly functioning IRs in the brain [31,32]. It was reported that, in the brains of aged mice, Akt was optimally phosphorylated by the infusion of insulin, not into peripheral blood, but into the cerebral ventricle. A reduced cerebrospinal fluid/serum insulin ratio was observed in an elderly human brain. This study concluded that insulin uptake at the BBB may be affected in the aged brain [28]. Another study reported that both brain insulin and c-peptide levels decreased alongside aging, and were lower in an AD brain than in an age-matched control. Moreover, decreased IR density was milder in an AD brain than in an age-matched control, suggesting that IR activity may be decreased in AD pathology [33]. In the hippocampal formation of an AD brain, the increased phosphorylation of the IRS was reported to be related to the deactivation of the IR-IRS-PI3K-Akt pathway. This observation was independent from the DM and APOE phenotype, and presented a negative correlation with Aβ accumulation, concluding that impaired function of IRs may be caused by Aβ-related IRS phosphorylation [31]. 

## 4. Declining Glucose Utilization and Preserving Ketone Metabolism in the Brain of AD

The decline in glucose metabolism is accompanied by the accumulation of Aβ. Aβ in cells directly damage mitochondria by attacking not only electron transport complex III [34], but also cytochrome c and many enzymes in the TCA cycle [1]. Moreover, Aβ produces reactive oxygen species (ROS) and reactive nitrogen species (RNS), which will damage the cell membrane, including glucose transporters and N-methyl-D-aspartate receptors [1]. These molecular reactions, induced by Aβ, may cause declining glucose utilization. In reality, Aβ deposition has been detected 15 years before the onset of AD symptoms, and cerebral hypometabolism has also been observed 10 years before the onset of AD [35,36]. 

A dual-tracer positron emission tomography (PET) study reported that the cerebral metabolic rate of glucose (CMRGlu) in AD patients was ~11% lower in the frontal, parietal, and temporal lobes in addition to the cingulate gyrus (*p* < 0.05), compared to healthy older adults. Moreover, the uptake rate constants of glucose (KGlu) in AD were ~15% lower in the same regions and subcortical regions than in healthy older adults [37]. Meanwhile, neither the regional nor whole-brain CMR of acetoacetate (CMRAcAc) and uptake rate constants of AcAc were significantly different between the healthy older adult controls and MCI or AD groups [37,38]. Additionally, it has been reported that CMRAcAc did not significantly differ between healthy young adults and healthy older adults [39]. These reports suggest that glucose metabolism easily declines when people get older or experience cognitive decline, while ketone metabolism is almost unchanged. 

Interventional studies focusing on a KD have reported that a 14-day high-fat KD using older rats showed a 28% and 44% increase in whole-brain CMRAcAc and CMRGlc, respectively [40,41]. On the other hand, a trial on human adults with a KD showed that the CMRAcAc was significantly increased (*p* = 0.005) but that the CMRGlu was significantly decreased by 20% (*p* = 0.014) [42]. Whether a KD increases or decreases glucose uptake in the brain after intervention remains an issue to be decided. 

## 5. KD, CO, and MCT Oil

As mentioned above, in contrast to the decline in cerebral glucose metabolism, since cerebral ketone metabolism is well preserved, a KD could be a potential intervention against decreased brain activity. 

Ordinary Americans consume 50–65% of their energy from carbohydrates, whereas a KD is a very high-fat, low-carbohydrate diet that restricts carbohydrates to ≤10% of the energy consumed. It is considered to be a difficult diet to maintain. This macronutrient profile promotes a systemic shift from glucose metabolism toward the conversion of fatty acids into ketone bodies as a substrate for energy. Some food products that are known to contain rich ketone body precursors are coconut oil and MCT oil. The coconut tree bears a large number of fruits in regions of India, Sri Lanka, Malaysia, and the Philippines. Coconut oil is derived from the pulp of coconut fruit and is used for cooking oil. Coconut oil is a source of polyphenols and is rich in dietary fiber, vitamins, and minerals [43]. Coconut oil is comprised of saturated fatty acids (SFAs) (>90%) and small amounts of mono- and poly-unsaturated fatty acids. In a randomized trial, coconut oil was shown to significantly increase high-density lipoprotein (HDL) levels compared to butter and olive oil [44,45]. The lipid profile of coconut oil was demonstrated to be better than that of animal oils and other plant oils [46,47]. Additionally, coconut oil has been reported to have numerous medicinal benefits, including antibacterial, antifungal, antiviral, antiparasitic, antidermatophytic, antioxidant, hypoglycemic, hepatoprotective, and immunostimulant effects [48]. Nonetheless, regular coconut oil intake should be cautiously considered because of its high levels of SFAs. Diets rich in SFAs are associated with coronary heart disease [49,50]. More research is needed to balance the risk of coconut oil in cardiovascular diseases against its other benefits. 

On the other hand, MCT oil consists only of MCTs, which are lipid molecules that are more readily absorbed and oxidized than most lipids. Medium-chain fatty acids (MCFAs) are saturated fatty acids with carbon chain lengths of C6, C8, C10, or C12. MCTs are comprised primarily of octanoic acid (C8), decanoic acid (C10), and small proportions of caprice acid (C6) and lauric acid (C12). In reality, 62–70% of SFAs become MCTs [51]. Therefore, it can be said that MCT oil is more effective than coconut oil, since coconut oil contains various kinds of SFAs. Dietary MCTs are partially hydrolyzed by lingual lipase in the stomach and then hydrolyzed rapidly and efficiently by pancreatic lipase within the intestinal lumen. A minor proportion of MCFAs bypass the liver and are distributed to peripheral tissues via general circulation [52,53,54]. Subsequently, MCFAs are directly absorbed through the gut via the portal vein to the liver, rather than through the thoracic duct lymph system, which is the conventional route for the absorption of triglycerides containing light-chain fatty acids [52]. Within the liver mitochondria, MCFAs are rapidly metabolized through β-oxidation and finally become ketone bodies, such as β-hydroxybutyrate (βHB), AcAc, and acetone. Thus, the unique characteristics of MCFAs, which are easily absorbed and metabolized, have led to interest in their use in the management of several gastrointestinal disorders, where MCTs have been primarily used to reduce fat malabsorption and serve as a source of calories to optimize their nutritional status.

## 6. Influence of a KD and MCT Oil on AD

The KD was originally developed as a treatment for epilepsy in the 1920s, as fasting was known to reduce the frequency of seizures [55]. Numerous studies have revealed the efficacy of a KD for epilepsy treatment [56,57,58]. 

On the other hand, a KD has been attractive as non-pharmacological treatment for T2DM. It was reported that, after 56 weeks of a KD, body weight, BMI, blood glucose level, total cholesterol, low-density lipoprotein cholesterol (LDL), triglycerides, and urea levels significantly decreased, whereas the level of HDL cholesterol significantly increased. Interestingly, these changes were more significant in subjects who had a high blood glucose level than in those with a normal blood glucose level [59]. Moreover, a KD was reported to show beneficial effects on weight loss in overweight and obese participants when compared to a conventional low-calorie diet [60]. 

By extension, the shift from glucose metabolism to ketone metabolism in patients with AD may be reasonable for neuronal protection. Patients with mild-to-moderate AD were assessed by neurocognitive tests after taking 50 g of a ketogenic formula containing 20 g of MCTs or an isocaloric placebo formula without MCTs [61]. The patients then took the ketogenic formula daily for up to 12 weeks and underwent neurocognitive tests monthly. At 8 weeks after the start of the trial, the patients showed a significant improvement in their immediate and delayed logical memory tests compared to their baseline scores; at 12 weeks, they showed significant improvements in the digit–symbol coding test and immediate logical memory test compared to the baseline. Hence, chronic consumption of the ketogenic formula has been suggested to have positive effects on verbal memory and processing speed in patients with AD. It was reported that AD patients with a KD who achieved sustained physiological ketosis showed an increase in mean within-individual AD cooperative study–activities of daily living (ADCS-ADL) (+3.13 ± 5.01 points, *p* = 0.0067) and quality of life in AD (+3.37 ± 6.86 points, *p* = 0.023) scores compared to those with a usual diet [62]. A meta-analysis of randomized controlled trials with 422 participants showed, compared with a placebo, a trend toward cognitive improvement on the AD assessment scale–cognitive subscale (ADAS-Cog) (MD = −0.539; 95% CI, −1.239–0.161, I2 = 0%), and significantly improved cognition when combining ADAS-Cog with the Mini-Mental State Examination (MMSE) (SMD = −0.289; 95% CI, −0.551–−0.027, I2 = 0%) [63]. Moreover, in frail elderly patients, MCT supplementation increased the MMSE score by 3.5 points at the 3-month intervention from baseline (*p* < 0.001), whereas long-chain triglyceride supplementation decreased the MMSE score by −0.7 points [64].

## 7. The Collateral Effects of MCT Oil for Cerebral Glucose Hypometabolism

Studies involving treating epilepsy patients with a KD have suggested that a KD introduces ketone bodies which influence the TCA cycle, membrane potential hyperpolarization, γ-aminobutyric acid synthesis, and decreasing glutamate release [65]. A KD may have many different effects as an anti-epileptic treatment. The possible benefit of a KD on T2DM patients is mainly because of the reduction in blood glucose level as well as improving the homeostasis model assessment of insulin resistance (HOMA-IR) [66]. In patients with AD, several mechanisms can be expected for the improvement of cognitive function regarding the introduction of a KD.

In brief, within the liver mitochondria, MCFAs are rapidly metabolized to ketone bodies, whereas the liver does not have enzymes that produce Ac-CoA from ketones [67,68], which means that the liver cannot produce energy from ketones via the TCA cycle. Ketones and minor MCFAs escape from liver metabolism and are transported to the whole body, especially the brain, where the enzymes produce Ac-CoA. Arriving at the brain, ketones and MCFAs can cross the BBB [69,70]; therefore, they enter the TCA cycle and generate energy. Ac-CoA is also generated from pyruvate, a glycolytic product, by pyruvate dehydrogenase. This can then enter the TCA cycle to generate more energy [71]. However, in AD patients, they cannot utilize sufficient glucose; thus, Ac-CoA, which comes through the glycolytic pathway, will lower its concentration in the mitochondria, resulting in a decrease in energy production. This mechanism may also be applied to patients with DM who cannot take glucose from the bloodstream. Ketone can bypass the blockade of glycolysis induced by insulin deficiency, thereby providing an alternative source of mitochondrial Ac-CoA [72]. AcAc and βHB, which are produced in the liver and supplied to the brain, may serve as alternative sources of energy in neurons. Through this pathway, a KD contributes to maintaining neuronal activity, leading to the improvement of cognitive performance. 

## 8. The Hypothesis of Direct Effects of MCT Oil (MCFAs) on Cognitive Performance

In addition to the role of ketones as energy sources for decreased glucose utilization in patients with AD, MCFAs may have other effects. Although there is insufficient firm evidence to support the hypothesis presented below, it is important to explore the possibility of other effects of MCT oil. 

### 8.1. Ligand for Peroxisome-Proliferator-Activated Receptor γ (PPARγ)

As mentioned above, Aβ attacks the mitochondria, whereas a KD may be involved in the maintenance of mitochondrial biogenesis (MB) and the mitochondrial respiratory chain (MRC). MB is controlled by nuclear sirtuins (SIRT1) [25,73], and a KD has been reported to improve energy metabolism and MB in the hippocampus of rats [74]. A KD may improve MB in neuronal cells, probably via PGC1-α and/or sirtuins [75]. It has been reported that when an HT22 mouse hippocampal neuronal cell line was incubated with decanoic acid, a main component of MCT oil, or βHB, a metabolite of MCFAs, a significant elevation of SIRT1 enzyme activity and an overall upregulation of MRC were observed [76]. In addition, decanoic acid was reported to function as a direct PPARγ ligand [77]. PPARγ is a subfamily of nuclear receptors that plays a significant role in glucose and lipid metabolism. In fact, it is a therapeutic target of the DM drug, thiazolidinedione [78]. Furthermore, PPARγ agonists were reported to promote the biogenesis of functional mitochondria [79]. One study attempted treatment with decanoic acid in individuals diagnosed with mitochondrial disease. This treatment increased citrate synthase activity, a marker of cellular mitochondrial content, in 50% of fibroblasts obtained from patients with Leigh syndrome [80]. Another study also showed that decanoic acid, but not octanoic acid, caused a marked increase in citrate synthase, along with complex I activity and catalase activity, in neuronal cell lines. They also observed an increase in mitochondrial number, as indicated by electron microscopy [81]. The other study reported that the HT22 mouse hippocampal neuronal cell line, incubated with decanoic acid, showed prominent increases in maximal activities of complexes I + III and complex IV of the MCR as well as ratios of their activities to that of citrate synthase [76].

### 8.2. Lactate Shuttle

As the lactate/pyruvate concentration ratio is significantly increased under hypoxia [82,83], lactate has long been considered a metabolic byproduct under hypoxic conditions. However, many recent studies have changed this view of lactate, shifting from a glycolytic waste product to an important energy fuel with an interesting molecular pathway [84,85]. It can be said that lactate produced by astrocytic glycolysis could be a supplementary fuel for neighboring neurons [86,87,88]. This is referred to as the astrocyte–neuron lactate shuttle (ANLS) hypothesis. As a consequence of high glycolytic activity, astrocytes release lactate into the extracellular space, which can then be taken up by neighboring neurons and serve as an oxidative fuel for their mitochondria [89,90]. One study showed that, using single-cell imaging, decanoic acid produced lactate by inducing glycolysis [89]. The researchers implied that decanoic acid works for astrocyte metabolism and supplies lactate to neighboring neurons by the ANLS. In the brain of patients with AD, this hypothesis remains controversial [84,91,92]. Since MCT oil contains decanoic acid, the effect of MCT oil through the ANLS may be an interesting target for the future.

## 9. Conclusions

In the brain of patients with AD, deficiencies of insulin utility and glucose metabolism are observed, similar to the pathophysiological connection to T2DM. In this pathological condition, not glucose but a ketone metabolism cascade could be an alternative pathway for protecting neurons. MCT oil, a critical component of a KD, is an interesting material for competing AD pathology, because it can be an effective source of ketone as well as maintaining mitochondrial function. Exploring the metabolic impairment in the brain of patients with AD will currently be important from the viewpoint of non-pharmaceutical therapy of AD.

## Figures and Tables

**Figure 1 ijms-22-12310-f001:**
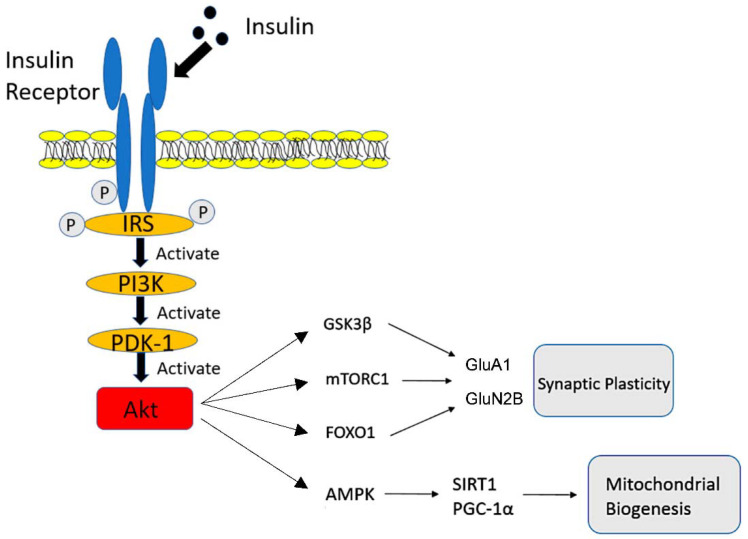
Implications of insulin in cognitive performance. Once an insulin receptor is activated, phosphorylation of the insulin receptor substrate (IRS) is triggered. Then, phosphoinositide 3-kinase (PI3K) and phosphoinositide-dependent protein kinase-1 (PDK-1) are activated, resulting in the activation of protein kinase B (Akt). Activated Akt promotes several downstream cascades and influences synaptic plasticity as well as mitochondrial dysfunction.

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
