# Peer review of "Alzheimer’s Disease and Type 2 Diabetes Mellitus: The Use of MCT Oil and a Ketogenic Diet"

_ijms, 2021, doi:10.3390/ijms222212310_

Round 1

Reviewer 1 Report

Takeishi and coworkers in their review describe the relationship between Alzheimer's disease and T2D. In addition, the authors discuss the impact of MCT oil and Ketogenic diet on AD and T2D. The present manuscript is a timely update however, some information needs to be elaborated before being further considered for publication.

Figure1: How does AKT activation results in synaptic plasticity should be elaborated in the text.

Authors should discuss the studies or hypotheses that discuss the impact of MCT oil and ketogenic diet on AD and T2D. Several studies discuss the individual relationship between diet-AD and diet and T2D.

Figure 3: the data presented in Figure 3 is still the primary data. It would be nice to have this manuscript in the form of a mini-review rather than a full review.

Reviewer 2 Report

The manuscript by Takeishi et al sets out to discuss the relationship between Type 2 Diabetes and Alzheimer’s Disease (AD), as well the potential therapeutic implications of medium-chain triglycerides and keto diet in preventing or slowing AD. The authors emphasized that the shift from glucose metabolism to ketone metabolism in patients with AD and DM can provide benefit due to the brains reduced glucose availability and utilization and the fact that ketone bodies can serve as an alternative carbon source from glucose, and that ketone metabolism is well preserved. While this topic is of significant interest to the neurodegeneration and diet-based therapeutic strategy fields, there are significant issues with the current manuscript that make it difficult to follow the logical flow of the topic being presented as well as the superficial nature of the discussion. Therefore, I believe in its current state, this manuscript is premature to accept for publication in International Journal of Molecular Sciences. Following are some suggestions that authors could utilize to improve or enhance the impact of this article.

Major concerns/questions:

  1. The manuscript often lacks logical flow and often glosses over important concepts of the topic in a superficial manner. The basis for the manuscript, the interrelationship between AD and T2D is often confusing. Both AD role in T2D as well as T2D role in AD are discussed but neither in a clear and concise manner. More emphasis should be placed on the manner and mechanisms by which T2D may predispose individuals to AD, and this should be detailed to a level that engages the reader in the topic and provides the rationale and background for the reader to follow the subsequent reasoning for MCT and KD as potential therapeutic strategies.
  2. Often details of clinical studies are provided at a level that is unnecessary (reminiscent of what an abstract of those studies might sound like). These studies should be summarized but don’t need to provide details of the data collected.
  3. Considering the controversial nature of aducanumab approval and whether it has significant efficacy, the discussion of this treatment in the introduction is somewhat premature, as well as not being fully relevant to the review, and thus should be left out.
  4. Section 2 is poorly written, lacks logical flow, and should be expanded to provide a more comprehensive description of mutations in key factors such as APP, PSEN1 and PSEN2, as well as other risk factors for AD.
  5. Reasoning for including primary data in figure 2 is unclear should be removed, given this is a review article.
  6. It is never clearly described why AD patients cannot utilize glucose. There are statements as to why the brain of AD patients have reduced insulin action, but the reasoning is stated as three possibilities, which are never discussed at length (but should be in this review).
  7. Ab attacking the mitochondria is superficially described.

Minor concerns/questions?

  1. The review Line 17: should it read “when the brain cannot…”
  2. A relationship between decanoic acid and coconut oil relation should be more clearly stated.
  3. First paragraph in page 2 is confusing as it is unclear at this point in the manuscript whether T2D predisposes individual to AD or whether AD pathology promotes T2D.
  4. Line 71, For example, the authors write “Ballard et al. extensively summarized…” It is more professional to remove the author reference and cite the reference at the end of the sentence.
  5. Line 228: The shift from glucose metabolism to ketone metabolism in patients with AD may be reasonable for neuronal protection. This statement should be provided also in the abstract.
  6. βH on line 302 is unclear as to what it represents.
  7. Line 325-326 seem out of place and not relevant to the lactate shuttle.

Round 2

Reviewer 1 Report

The authors have put a lot of effort to improve their manuscript. The manuscript looks much improved upon revision. There are only a few typographical and grammatical errors that need to be addressed before final publication. The take-home message of the manuscript is clear now and the flow of information is more logical and easy to read. The manuscript is well within the scope of the journal and may be accepted for publication. I congratulate the team for their good work. 

Reviewer 2 Report

The manuscript by Takeishi et al sets out to discuss the relationship between Type 2 Diabetes and Alzheimer’s Disease (AD), as well the potential therapeutic implications of medium-chain triglycerides and keto diet in preventing or slowing AD. The authors emphasized that the shift from glucose metabolism to ketone metabolism in patients with AD and DM can provide benefit due to the brains reduced glucose availability and utilization and the fact that ketone bodies can serve as an alternative carbon source from glucose, and that ketone metabolism is well preserved. This topic is of significant interest to the neurodegeneration and diet-based therapeutic strategy fields. This revision is much improved over the initial submission. I have few additional suggestions that the authors should consider to enhance its readability and to provide a more solid framework on the interrelationship between T2DM and AD.

Major concerns/questions:

  • The authors have better explained the connection between AD and impaired insulin action than the in the initial submission. I still find that the cause-and-effect relationship between Alzheimer's and diabetes may not be easy to make for the casual reader.

Minor concerns/questions

The English grammar is much improved, I have listed remaining issues below:

  • Line 34, the authors may want to choose a different word instead of “spoil”.
  • Line 36 should read “Regarding glucose metabolism disorders,”
  • Line 95, “in the brain of aged mouse” should be changed to “in brains of aged mice”
  • First line of section 7 should read, “Studies involving treating epilepsy patients with KD, it…
  • Line 289, it should start “In the AD brain,…”
